# Clinical Characteristics and Predictors of Mortality in Critically Ill Adult Patients with Influenza Infection

**DOI:** 10.3390/ijerph18073682

**Published:** 2021-04-01

**Authors:** Wei-Cheng Hong, Shu-Fen Sun, Chien-Wei Hsu, David-Lin Lee, Chao-Hsien Lee

**Affiliations:** 1Division of Pulmonary Medicine, Kaohsiung Veterans General Hospital, Kaohsiung 813779, Taiwan; george61394@gmail.com (W.-C.H.); llee@vghks.gov.tw (D.-L.L.); 2School of Medicine, National Yang-Ming University, Taipei 112304, Taiwan; sfsun@vghks.gov.tw; 3Department of Health Business Administration, Meiho University, Pingtung 912009, Taiwan; x00002167@meiho.edu.tw

**Keywords:** acute respiratory distress syndrome, extracorporeal membrane oxygenation, multiple organ failure, outcome, severe influenza

## Abstract

Patients with influenza infection may develop acute respiratory distress syndrome (ARDS), which is associated with high mortality. Some patients with ARDS receiving extracorporeal membrane oxygenation (ECMO) support die of infectious complications. We aimed to investigate the risk factors affecting the clinical outcomes in critically ill patients with influenza. We retrospectively reviewed the medical records of influenza patients between January 2006 and May 2016 at the Kaohsiung Veterans General Hospital in Taiwan. Patients aged below 20 years or without laboratory-confirmed influenza were excluded. Critically ill patients who presented with ARDS (*P* = 0.004, odds ratio (OR): 8.054, 95% confidence interval (CI): 1.975–32.855), a higher Acute Physiology and Chronic Health Evaluation (APACHE) II score (*P* = 0.008, OR: 1.102, 95% CI: 1.025–1.184), or higher positive end-expiratory pressure (*P* = 0.008, OR: 1.259, 95% CI: 1.061–1.493) may have a higher risk of receiving ECMO. Influenza A (*P* = 0.037, OR: 0.105, 95% CI: 0.013–0.876) and multiple organ failure (*P* = 0.007, OR: 0.056, 95% CI: 0.007–0.457) were significantly associated with higher mortality rates. In conclusion, our study showed critically ill influenza patients with ARDS, higher APACHE II scores, and higher positive end-expiratory pressure have a higher risk of receiving ECMO support. Influenza A and multiple organ failure are predictors of mortality.

## 1. Introduction

Influenza viruses have segmented genomes to enable virus reassortments and display antigenic variation [1]. There are three types of influenza viruses: A, B, and C [1]. However, only types A and B have caused widespread outbreaks [1]. The H1N1 pandemic from 1918 to 1919 caused 40–50 million deaths globally [1]. Over 60.8 million cases, 274,304 hospitalizations, and 12,469 deaths were reported in the entire United States due to the 2009 influenza A virus (pH1N1) pandemic from 12 April 2009 to 10 April 2010 [2]. The clinical symptoms of human influenza virus infection are sudden fever 1–3 days following infection, headache, limb ache, tiredness, general faintness, and dry cough [3]. Influenza infections are mostly self-limiting, but some patients may develop severe complications, such as pneumonia or acute respiratory distress syndrome (ARDS) [4]. Some studies have demonstrated that the risk factors for severe influenza infection include age, extremes of age, pregnancy, chronic underlying medical conditions, being a resident in a nursing home, obesity, and young obese but previously healthy persons [5,6,7]. The use of corticosteroids has been associated with poor prognosis in patients with influenza pneumonia [8]. H1N1-ARDS progresses and renders patients prone to life-threatening hypoxemia [9]. H1N1-ARDS has a substantially different clinical course from non-H1N1-ARDS, which shows a prolonged recovery of pulmonary gas exchange, the demand for extracorporeal lung support growth, and a protracted intensive care unit (ICU) stay [9]. Extracorporeal membrane oxygenation (ECMO) is a rescue treatment for ARDS [10]. However, some patients with ARDS receiving ECMO support die of infectious complications, leading to multiple organ failure [11]. This study aimed to investigate the clinical characteristics and risk factors affecting the clinical outcomes of critically ill patients with influenza.

## 2. Materials and Methods

We retrospectively reviewed the medical records of adult patients with laboratory-confirmed severe influenza infection admitted in the ICU between January 2006 and May 2016 at the Kaohsiung Veterans General Hospital in Taiwan. Patients aged below 20 years and without laboratory-confirmed influenza infection were excluded. Influenza was diagnosed based on the following criteria: positive for rapid influenza diagnostic test, reverse-transcription polymerase chain reaction, or virus culture in the respiratory specimen.

We used electronic medical records and supplemental manual records of patients from the hospital. The study was approved by the Institutional Review Board of Kaohsiung Veterans General Hospital (IRB: VGHKS16-CT8-10). The need for informed consent was waived due to the retrospective nature of this study, and the patients’ data were anonymized before analysis. The medical records of this cohort were assessed to determine the patients’ sex, age, body mass index, comorbidity, influenza type, organ failure number, presence of ARDS, Acute Physiology and Chronic Health Evaluation (APACHE) II score, and use of steroids or ECMO. Respiratory parameters included arterial oxygen tension/fraction of inspired oxygen ratio (P/F) ratio, positive end-expiratory pressure (PEEP), lung compliance, duration of mechanical ventilation, ICU and hospital stay, and mortality.

We analyzed the comorbidities such as hypertension and diabetes mellitus. Multiple organ failure was defined as the failure of two or more organs, and acute respiratory failure was defined as requiring endotracheal tube intubation and mechanical ventilation, or the use of noninvasive positive ventilation. According to the Berlin criteria, ARDS was defined as acute respiratory distress characterized by bilateral opacities on chest radiographs consistent with pulmonary edema within 1 week of known clinical insult, not fully explained by effusions, lobar/lung collapse, or nodules with severe hypoxemia (P/F ratio < 300 mmHg) in the absence of cardiogenic pulmonary edema [12]. Lung compliance was defined as the ratio of tidal volume to positive inspiratory pressure-positive end-expiratory pressure. All patients followed the same protocol, based on the sepsis campaign guidelines, with ARDS treated with protective lung strategy, including a high PEEP (≥5 cm H_2_O), low tidal volume (4–8 mL/kg), plateau pressure (<30 cm H_2_O) [13], and early goal-directed therapy for shock patients [14]. Other general medical treatments, including the use of sedatives, antibiotics, and paralysis with atracurium, were decided by the doctor in the ICU. We used the APACHE II score to evaluate the severity of the patient’s condition within 24 h of ICU admission. Mortality was defined as all-cause mortality during hospitalization.

### Statistical Analysis

We used the Statistical Package for the Social Sciences (SPSS, Inc., Chicago, IL, USA) version 25.0 to analyze the data. According to the patient characteristics, continuous variables were expressed as the mean ± standard deviation, while discrete variables were expressed as counts (percentage). To analyze the predictors of ECMO use and outcomes of severe influenza, we initially compared the demographic data, clinical characteristics, and in-hospital complications of survivors, nonsurvivors, and ECMO and non-ECMO users using a Student’s *t*-test for continuous variables and chi-square test for categorical variables. The optimal logistic regression model was selected using a stepwise model selection method. A *p* value of 0.05 or less was considered significant.

## 3. Results

### 3.1. Demographic Data

From January 2006 to May 2016, 99 critically ill patients with influenza (61 men and 38 women) were included, with a mean age of 62.2 ± 16.8 years and a mean BMI of 25.7 ± 6.0 kg/m^2^. Thirty-eight (38.4%) patients had diabetes mellitus, while 56 (56.6%) had hypertension; 83 (83.8%) of the patients had influenza A, including 25 (25.3%) with H1N1. The mean APACHE II score for assessing disease severity was 22.5 ± 8.5. Thirty-nine (39.4%) patients developed ARDS, 73 (73.7%) had multiple organ failure, 82 (82.8%) used steroids, and 20 used EMCO. The mean arterial oxygen partial pressure (PaO_2_)/fractional inspired oxygen (FiO_2_), PEEP, and lung compliance were 184.2 ± 161.7 cm H_2_O, 7.4 ± 3.4 cm H_2_O, and 27.9 ± 12.2 mL/cm H_2_O, respectively. The mean durations of mechanical ventilation use, length of ICU stay, and length of hospital stay were 19.1 ± 17.5, 12.9 ± 11.5, and 30.0 ± 23.3, respectively. The mortality rate was 30.3% (30/99). The demographic and clinical characteristics of critically ill patients with influenza are shown in Table 1.

Comparison between patients on ECMO and without ECMO support.

In comparison with patients without ECMO support, those on ECMO support had a relatively younger mean age (51.0 ± 15.2 vs. 65.0 ± 16.1 years, *P* = 0.001), higher incidence of H1N1 (50% vs. 19%, *P* = 0.004), multiple organ failure (100% vs. 67.1%, *P* = 0.003), ARDS (80% vs. 29.1%, *P* < 0.001), higher mean APACHE II score (26.8 ± 9.9 vs. 21.3 ± 7.8 points, *P* = 0.011), lower mean PaO_2_/FiO_2_ ratio (92.9 ± 96.4 vs. 207.9 ± 167.2 mL/cm H_2_O, *P* = 0.004), and higher mean PEEP (10.1 ± 4.4 vs. 6.6 ± 2.7 cm H_2_O, *P* < 0.001), as shown in Table 2.

In the multivariable logistic regression analysis of the predictors of ECMO use, critically ill patients with influenza who presented with ARDS (*P* = 0.004, odds ratio (OR): 8.054, 95% confidence interval (CI): 1.975–32.855), a higher APACHE II score (*P* = 0.008, OR: 1.102, 95% CI: 1.025–1.184), or higher PEEP (*P* = 0.008, OR: 1.259, 95% CI: 1.061–1.493) had a higher risk of receiving ECMO support (Table 3).

Patients under ECMO support had a longer duration of mechanical ventilation (26.3 ± 20.6 vs. 17.3 ± 16.2 days, *P* = 0.038) and ICU stay (20.5 ± 14.0 vs. 11.0 ± 10.0 days, *P* = 0.001) than those without, as shown in Table 2.

### 3.2. Predictor for Mortality

As shown in Table 4, a significant number of critically ill patients with influenza died due to influenza A (96.7% vs. 78.3%, *P* = 0.022), multiple organ failure (96.7% vs. 63.8%, *P* = 0.001), and ARDS (56.7% vs. 32.4%, *P* = 0.023). Multivariable logistic regression analysis of the predictors of mortality demonstrated a significant association of influenza A (*P* = 0.037, OR = 0.105, 95% CI: 0.013–0.876) and multiple organ failure (*P* = 0.007, OR: 0.056, 95% CI: 0.007–0.457) with a higher mortality rate in critically ill patients with influenza infection, as displayed in Table 5.

## 4. Discussion

In the present study, ARDS complicated by influenza was observed in 39.4% of critically ill patients with influenza, which was similar to the results of previous studies (39.6% in 33 hospitals in the United States [15], and an ARDS incidence of 38% was observed among severely ill patients in the 2009 pandemic year [16]). In a CESAR study, ECMO support significantly improved the survival rate of patients with severe acute respiratory failure [17]. In patients with ARDS, ECMO is a rescue treatment; however, a high sequential organ failure assessment score before ECMO demonstrated a lower survival rate [11]. A higher survival rate was observed in patients with severe influenza on ECMO support without multiple organ failure than in those who developed multiple organ failure [18]. The present study findings showed that ECMO use was not associated with higher survival rates in critically ill patients with influenza; however, it may be related to only 39.4% of the total patients who developed ARDS. Critically ill patients with influenza on ECMO support had a higher APACHE II score, indicating a high severity of disease in the ECMO group. Hence, patients receiving ECMO support did not reduce the risk of mortality. There were 39 critically ill influenza patients who developed ARDS, with 16 patients receiving and 23 not receiving ECMO support. However, due to the small sample size, further studies are warranted to evaluate the predictors of mortality and benefits of using ECMO in critically ill patients with influenza who develop ARDS.

Influenza A is associated with a high mortality rate [19]; however, this association was not significant according to a systematic literature review [20], while another study showed that influenza A was associated with a higher rate of disease severity than H3N2 and influenza B [21]. In our study, influenza A was significantly associated with a higher mortality rate in critically ill patients with influenza (*P* = 0.037, OR = 0.105, 95% CI: 0.013–0.876).

Pulmonary fibrosis is a severe complication in post-ARDS patients with influenza [22], which may prolong mechanical ventilation use and ECMO support and cause poor lung function after recovery. In this study, the effect of poor lung compliance in patients with severe influenza showed a relative risk trend for mortality; however, no significant difference was observed in the mortality rate (*P* = 0.06, OR: 1.049, 95% CI: 0.998–1.102) according to the multivariable logistic regression analysis. Further studies can be conducted to evaluate the association between the effects of lung compliance and quality of life in critically ill patients with influenza after their recovery.

Meduri et al. reported the benefit of steroids in patients with ARDS in a randomized controlled trial in significantly improving pulmonary and extrapulmonary organ dysfunction and reducing the duration of mechanical ventilation and ICU stay [23]. In another study, prolonged glucocorticoid treatment was associated with a substantial and significant improvement in meaningful patient-centered outcome variables and a distinct survival benefit when treatment was initiated before Day 14 of ARDS [24]. However, early use of corticosteroids in patients affected by the (H1N1)v influenza A infection pandemic did not result in better outcomes and was associated with an increased risk of superinfections [25]. One systematic review and meta-analysis reported that corticosteroid use was associated with higher mortality in patients with influenza pneumonia [8]. In this study, the use of steroids was not a predictor for mortality, and this may be related to the following clinical characteristics: shock (82 patients), chronic obstructive pulmonary disease (9 patients), asthma (6 patients), systemic lupus erythematosus (1 patient), and rheumatoid arthritis (1 patient), which required steroids as treatment.

In 33 American hospitals, in 2013–2014, 19.1% of patients with severe influenza had died [15]. Vandroux et al. reported the death rate of 31% of the 127 patients with severe influenza admitted in an ICU in Reunion Island [26]. In another study, 40% of the critically ill patients with influenza admitted to an ICU in the Netherlands died [27]. In the present study, the high mortality rate of 30.3% could be attributed to the high severity of disease, high APACHE II score (estimated mortality rate of 40% with an APACHE II score of 22.5 ± 8.5), and a high proportion of multiple organ failure (73.7%).

This study has several limitations. First, as a retrospective study, data on vaccination status (influenza vaccination and pneumococcus vaccination) needed for analysis were missing. Second, we excluded the population aged below 20 years. A study in Spain showed that severe influenza affected younger patients, especially young obese but previously healthy ones, causing pulmonary complications [6]. Therefore, the results of our study cannot be applied to the general population. Third, younger patients may have a tendency to accept ECMO support than older patients, and financial issues are also a consideration. Finally, the number of patients included in this study was relatively small.

## 5. Conclusions

In the small cohort study, we identified the characteristics of critically ill influenza patients and found that ARDS, a higher APACHE II score, and a higher PEEP increased the risk of receiving ECMO support. Moreover, influenza A and multiple organ failure were independently associated with higher mortality rates in critically ill influenza patients.

## Figures and Tables

**Table 1 ijerph-18-03682-t001:** Demographic and clinical characteristics of critically ill patients with influenza.

	Total Patients (*n* = 99)
Demographic	
Gender (M/F)	61/38
Age (years)	62.2 ± 16.8
BMI	25.7 ± 6.0
Underlying disease	
Diabetes mellitus	38 (38.4)
Hypertension	56 (56.6)
Heart failure	11 (11.1)
End-stage renal disease	3 (3.0)
COPD	9 (9.1)
Asthma	6 (6.1)
SLE	1 (1.0)
RA	1 (1.0)
Causative agents	
Influenza A	83 (83.8)
Non-influenza A	16 (16.2)
H1N1	25 (25.3)
Comorbidities	
Pneumothorax	6 (6.1)
Number of multiple organ failure	73 (73.7)
ARDS	39 (39.4)
APACHE II score	22.5 ± 8.5
Shock	82 (82.8)
Treatment	
Use of steroid	82 (82.8)
Respiratory parameters	
PaO_2_/ FiO_2_ (mm Hg)	184.2 ± 161.7
PEEP (cm H_2_O)	7.4 ± 3.4
Compliance (mL/cm H_2_O)	27.9 ± 12.2
Outcome	
Duration of mechanical ventilation (days)	19.1 ± 17.5
Length of ICU stay (days)	12.9 ± 11.5
Length of hospital stay (days)	30.0 ± 23.3
Mortality	30 (30.3)

BMI, body mass index; COPD, chronic obstructive pulmonary disease; SLE, systemic lupus erythematosus; RA, rheumatoid arthritis, ARDS, acute respiratory distress syndrome; APACHE, Acute Physiology and Chronic Health Evaluation; PaO_2_/ FiO_2_, arterial oxygen tension/fraction of inspired oxygen; PEEP, positive end-expiratory pressure; ICU, intensive care unit. All values are expressed as the number of patients (%) or mean ± standard deviation.

**Table 2 ijerph-18-03682-t002:** Differences in the demographic and clinical characteristics between critically ill patients with influenza on ECMO and those on non-ECMO support.

	ECMO (*n* = 20)	Non-ECMO (*n* = 79)	*p* Value
Demographic			
Gender (M/F)	14–6	47/32	0.388
Age (years)	51.0 ± 15.2	65.0 ± 16.1	0.001
BMI	28.0 ± 7.6	25.1 ± 5.5	0.058
Underlying disease			
Diabetes mellitus	6 (30.0)	32 (40.5)	0.388
Hypertension	7 (35.0)	49 (62.0)	0.029
Causative agents			
Influenza A	17 (85.0)	66 (83.5)	0.874
Non-influenza A	3 (15.0)	13 (16.4)	0.874
H1N1	10 (50.0)	15 (19.0)	0.004
Comorbidities			
Pneumothorax	2 (10.0)	4 (5.1)	0.408
Number of multiple organ failure	20 (100)	53 (67.1)	0.003
ARDS	16 (80.0)	23 (29.1)	<0.001
APACHE II score	26.8 ± 9.9	21.3 ± 7.8	0.011
Treatment			
Use of steroid	17 (85.5)	66 (83.5)	0.874
Respiratory parameters			
PaO_2_/FiO_2_ (mm Hg)	92.9 ± 96.4	207.9 ± 167.2	0.004
PEEP (cm H_2_O)	10.1 ± 4.4	6.6 ± 2.7	<0.001
Compliance (mL/cm H_2_O)	28.1 ± 9.0	28.0 ± 13.0	0.917
Outcome			
Duration of mechanical ventilation (days)	26.3 ± 20.6	17.3 ± 16.2	0.038
Length of ICU stay (days)	20.5 ± 14.0	11.0 ± 10.0	0.001
Length of hospital stay (days)	32.7 ± 24.9	29.3 ± 23.0	0.565
Mortality	9 (45.0)	21 (26.6)	0.109

ECMO, extracorporeal membrane oxygenation; BMI, body mass index; ARDS, acute respiratory distress syndrome; APACHE, Acute Physiology and Chronic Health Evaluation; PaO_2_/FiO_2_, arterial oxygen tension/fraction of inspired oxygen; PEEP, positive end-expiratory pressure; ICU, intensive care unit. All values are expressed as the number of patients (%) or mean ± standard deviation.

**Table 3 ijerph-18-03682-t003:** Multivariable logistic regression analysis of the predictors of ECMO use in critically ill patients with influenza.

Variable	*p* Value	Odd Ratio (95% C.I.)
ARDS	0.004	8.054 (1.975–32.855)
APACHE II Score	0.008	1.102 (1.025–1.184)
PEEP	0.008	1.259 (1.061–1.493)

ECMO, extracorporeal membrane oxygenation; ARDS, acute respiratory distress syndrome; APACHE, Acute Physiology and Chronic Health Evaluation; PEEP, positive end-expiratory pressure.

**Table 4 ijerph-18-03682-t004:** Demographic and clinical characteristics of survivors and nonsurvivors in critically ill patients with influenza.

	Survivors (*n* = 69)	Nonsurvivors (*n* = 30)	*p* Value
Demographic			
Gender (M/F)	46/23	15/15	0.117
Age (years)	61.2 ± 17.2	64.6 ± 16.0	0.484
BMI	25.8 ± 6.0	25.6 ± 6.4	0.854
Underlying disease			
Diabetes mellitus	29 (42.0)	9 (30.0)	0.258
Hypertension	39 (56.5)	17 (56.7)	0.989
Causative agents			
Influenza A	54 (78.3)	29 (96.7)	0.022
Non-influenza A	15 (21.7)	1 (3.3)	0.022
H1N1	16 (23.0)	9 (30.0)	0.473
Comorbidities			
Pneumothorax	4 (5.8)	2 (6.7)	0.868
Number of multiple organ failure	44 (63.8)	29 (96.7)	0.001
ARDS	22 (32.4)	17 (56.7)	0.023
APACHE II score	21.1 ± 7.5	25.6 ± 9.9	0.097
Use of ECMO	11 (15.9)	9 (30.0)	0.109
Treatment			
Use of steroid	59 (85.5)	24 (80.0)	0.494
Respiratory parameters			
PaO_2_/FiO_2_ (mm Hg)	195.5 ± 169.7	157.7 ± 140.6	0.380
PEEP (cm H_2_O)	7.4 ± 3.5	7.4 ± 3.5	0.932
Compliance (mL/cm H_2_O)	29.1 ± 13.2	24.8 ± 8.9	0.146
Outcome			
Duration of mechanical ventilation (days)	21.3 ± 17.5	14.1 ± 16.4	0.437
Length of ICU stay (days)	14.3 ± 11.6	9.6 ± 10.9	0.672
Length of hospital stay (days)	33.7 ± 21.7	21.4 ± 24.9	0.820

BMI, body mass index; ARDS, acute respiratory distress syndrome; APACHE, Acute Physiology and Chronic Health Evaluation; ECMO, extracorporeal membrane oxygenation; PaO_2_/ FiO_2_, arterial oxygen tension/fraction of inspired oxygen; PEEP, positive end-expiratory pressure; ICU, intensive care unit. All values are expressed as the number of patients (%) or mean ± standard deviation.

**Table 5 ijerph-18-03682-t005:** Multivariable logistic regression analysis for predictor of mortality in critically ill patients with influenza.

Variable	*p* Value	Odd Ratio (95% C.I.)
Influenza A	0.037	0.105 (0.013–0.876)
Multiple organ failure	0.007	0.056 (0.007–0.457)
Lung compliance	0.06	1.049 (0.998–1.102)

## Data Availability

All data relevant to the study are included in the article.

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
