# Peer review of "Clinical Characteristics and Predictors of Mortality in Critically Ill Adult Patients with Influenza Infection"

_ijerph, 2021, doi:10.3390/ijerph18073682_

Round 1
Reviewer 1 Report
I would like to congratulate you on your hard work and putting together this piece of work , I appreciate your honesty where you clearly mentioned about the limitation of your study and small sample size. It should have been also mentioned that results of this study and its conclusion should be interpreted cautiously.
Reviewer 2 Report
Well written and interesting article.
Despite the small study cohort, this work could give rise to other insights.
(1). In any case, it would have been interesting to relate the number of ARDS patients who then benefited from ECMO support and their mortality, perhaps with a summary table.
(2). The data on cortisone treatment fades into the background or is not well developed.
(3). It didn't seem to fit in perfectly with the final message of the work
Reviewer 3 Report
Thank you to the authors of the manuscript entitled " Clinical characteristics and predictors of mortality in adult critically ill patients with influenza infection” that have put together a considerable amount of work in a single-center retrospective study. They have tried to investigate the risk factors affecting clinical outcome in critically ill patients with influenza. However, there are a few challenges that need to be addressed. The principal thesis of this manuscript, which does not appear to be novel, had been discussed in several published studies (Sarda C et al,2019; Duan J et al,2020)
It is appreciable that the manuscript provides:
- Highlight the novelty of the study findings.
- Provide, previous vaccination of the patients.
- Please document that the different treatment regime of the patients didn’t affect the study outcome.
- One of the findings in the study showed that ECMO use was not associated with higher survival rates in critically ill patients with influenza .This finding is in conflict with other published studies (Abaziou T et al,2020).
- Reorganize the tables in subcategories (demographic, causative agents, underlying disease and comorbidities, treatment and hemodynamics, and outcome) .
- Several sentences have been structured incorrectly. Therefore, langue editing is required.
Round 2
Reviewer 3 Report
Thank you to the authors addressing the comments.
This revised version of the manuscript provides amelioration with respect to the original submission .